# Tribological and Mechanical Behavior of Graphite Composites of Polytetrafluoroethylene (PTFE) Irradiated by the Electron Beam

**DOI:** 10.3390/polym12081676

**Published:** 2020-07-28

**Authors:** Adrian Barylski, Andrzej S. Swinarew, Krzysztof Aniołek, Sławomir Kaptacz, Jadwiga Gabor, Arkadiusz Stanula, Zbigniew Waśkiewicz, Beat Knechtle

**Affiliations:** 1Faculty of Science and Technology, Institute of Materials Engineering, University of Silesia in Katowice, 75 Pułku Piechoty 1A, 41-500 Chorzów, Poland; adrian.barylski@us.edu.pl (A.B.); andrzej.swinarew@us.edu.pl (A.S.S.); krzysztof.aniolek@us.edu.pl (K.A.); slawomir.kaptacz@us.edu.pl (S.K.); jadwiga.gabor@us.edu.pl (J.G.); 2Institute of Sport Science, The Jerzy Kukuczka Academy of Physical Education, Mikołowska 72A, 40-065 Katowice, Poland; a.stanula@awf.katowice.pl (A.S.); z.waskiewicz@awf.katowice.pl (Z.W.); 3Department of Sports Medicine and Medical Rehabilitation, Sechenov University, 119991 Moscow, Russia; 4Institute of Primary Care, University of Zurich, 8091 Zurich, Switzerland

**Keywords:** PTFE, graphite, electron beam irradiation, differential scanning calorimetry (DSC), mechanical properties, wear

## Abstract

This research investigated the effect of irradiation with an electron beam energy of 10 MeV in doses of 26–156 kGy on polytetrafluoroethylene (PTFE) with a 15% and 20% graphite additive. The research has shown that mechanical (compression strength, hardness, and Young’s modulus) and sclerometric (coefficient of wear micromechanism and coefficient of resistance to wear) properties improve and tribological wear decreases as graphite content increases. Electron beam irradiation increases the degree of crystallinity of both materials to a similar extent. However significant differences in the improvement of all examined properties have been demonstrated for PTFE with higher (20%) graphite content subjected to the electron beam irradiation. This polymer is characterized by higher hardness and Young’s modulus, reduced susceptibility to permanent deformation, higher elasticity, compression strength, and above all, a nearly 30% reduction in tribological wear compared to PTFE with a 15% graphite additive. The most advantageous properties can be obtained for both of the examined composites after absorbing a dose of 104 kGy. The obtained results hold promise for the improvement of the operational life of friction couples which do not require lubrication, used for example in air compressors and engines, and for the possibility of application of these modified polymers. In particular PTFE with 20% graphite content, in the nuclear and space industry.

## 1. Introduction

Polytetrafluoroethylene (PTFE) has favorable properties like high chemical and biological tolerance, thermal stability, as well as its excellent dielectric, antifriction, and antiadhesive properties, polytetrafluoroethylene (PTFE) is a preferable material for the manufacture of articles in different industries and engineering [1,2,3,4,5,6]. For tribological applications, PTFE is also an expedient polymer material that is widely available. The structure of PTFE molecules causes transfer of material thin film onto the surface during friction improving value of the friction coefficient [7,8,9]. It is a known fact that during tribological interaction pure PTFE without technological defects resulting from the manufacturing process like polymer plates and nodules (few tenths of a millimeter) is featured by a very high degree of wear [7]. There are several ways to improve the wear rate, for example by adding different types of fillers to the material and also by using electron-beam or gamma-beam irradiation [9,10,11,12]. For example, graphite additive has a positive effect on parameters such as friction coefficient, stiffness, and mechanical strength, and it also slightly improves abrasion wear values [7].

First attempts to use ionizing radiation to modify polytetrafluoroethylene were already made in the sixties. [13,14]. At first, these attempts were ineffective because even with very low doses they caused deterioration of physical and mechanical properties and because of that PTFE was classified as one of polymers with the highest sensitivity to irradiation [15,16].

Subsequent research has shown, however, that irradiation improves a number of mechanical properties such as abrasion resistance, creep resistance, adhesive strength, abrasion strength, and also have positive influence on radiation resistance, processability, possibility for modification and enables control over properties of the obtained material [17]. An increase in resistance to abrasive wear of irradiated at ambient temperature PTFE was observed [18,19,20]. The initial failures were caused by the fact that in PTFE at room temperature the predominant phenomenon is the breakage of the main polymer chain, and the cross-linking phenomenon occurs only at a temperature similar to the melting point of PTFE [17,18,19,20]. Tabata et al. proved that the cross-linking of PTFE with a linear structure may occur only in type Y because the steric hindrance of geometric conformation prevents the formation of H type cross-linking [17]. The detailed mechanism that occur during the cross-linking of polytetrafluoroethylene have been described in detail in Khatipov’s paper [21]. Khatipov’s team also investigated friction and abrasion properties of pure PTFE and PTFE irradiated with doses up to 350 kGy. It has been demonstrated that the friction coefficient of specimens irradiated at a high temperature was close to the value of the material in its initial state, but its wear decreased by three–four times. Furthermore, hardly any wear products have been observed during the friction process, which may be very important for application of PTFE in so-called pure technologies, e.g., in the production of semi-conductors [22].

Current research is focused on irradiation of pure polytetrafluoroethylene with gamma rays due to the availability of this type of source, while sources enabling electron radiation with sufficient energy to penetrate the material to a depth of few cm are less common [20]. In the previous paper by these authors [23,24], results similar to Khatipov’s team data were obtained for a high-energy beam of electrons at room temperature. The aim of this study was to investigate the result of an increase in graphite content to 20% and radiation with a 10 MeV electron beam energy on mechanical, sclerometic and tribological properties of modified PTFE. The possibility of improving in this manner the properties of polymer at ambient temperature would decrease costs related to polytetrafluoroethylene modifications.

## 2. Materials and Methods

The research material was polytetrafluoroethylene with 15% and 20% graphite filling (SM-G15 and SM-G20, Inbras, Tarnów, Poland). The rods were made by sintering from PTFE suspension grounded and granulated. The additive used was electrographite with a maximum grain size of 63 µm, a minimum carbon content of 95%, including 80% in the graphite structure. Test specimens were cut out in the form of pins with a diameter of 5 mm (tribological tests) and cylinders with a diameter of 20 mm (other tests). The finished material was subjected to irradiation with an electron beam on an accelerator with an energy of 10 MeV and a power of 10 kW (NPO Torij, Moscow, Russia). Dosages from 26–156 kGy (2.6–15.6 Mrad) were used. The process was carried out at room temperature in a vacuum. Then the samples were annealed at 200 °C for 4 h and slowly cooled to room temperature.

### 2.1. Examination of Thermal Properties of PTFE-Graphite Composites

The degree of crystallinity of the tested composites was determined by differential scanning calorimetry (DSC) on a Mettler-Toledo DSC 1 device (Mettler-Toledo GmbH, Greifensee, Switzerland). Samples were about 0.015 g and they were cut from the central part of the cylinder and sealed in aluminum cells. The heating rate was 0.167 °C/S. Thermograms were registered for the melting process at temperatures from −40 to 400 °C, and for the recrystallization process from 400 °C to −40 °C. Registered data allowed to determine the degree of crystallinity χ_c_, using Formula (1) [25]. The tests were carried out on polymers in the initial state and after exposure to an electron beam.
(1)$$\chi_{c}=\frac{{\Delta H}_{c}}{{\Delta H}_{f}}\cdot 100\;\left[\%\right]$$
where: ΔH_c_—heat of phase transition from a DSC thermogram [cal/g]; ΔH_f_—heat of crystal phase transition of PTFE (19,585 cal/g).

The heat ΔH_c_, using Formula (2), allows one to calculate the number average molecular weight M_n_ of polytetrafluoroethylene-graphite composites [26]:(2)$$M_{n}=2.1\cdot 10^{10}\cdot {\Delta H}_{c}^{-5.16}$$

### 2.2. Compressive Strength Tests of PTFE-Graphite Composites

Uniaxial compression of polytetrafluoroethylene composites with 15% and 20% graphite content was performed on Instron 5982, with a constant deformation rate of 4.2 × 10^−3^ s^−1^. The deformation was performed up to half the height of the samples, then the system was unloaded without supporting and the maximum stress was determined σ_max_ = P_max_/A_0_, where A_0_ is the area of the initial cross-section of the tested samples, and P_max_ is the maximum force recorded at 50% compression. The measurements were performed at a temperature of 21 ± 1 °C, for each tested variant 5 repetitions were performed. 

### 2.3. Tests of Micromechanical Properties

Microindentation tests were performed on a Micron-Gamma device (manufactured by the Aviation Faculty, Technical University of Kiev, Kiev, Ukraine). For measurements, a Berkovich indenter with a pyramid shape and an angle between the center line and each wall equal to 65.3° was used. Measurement parameters were as follows: maximum load—1 N and holding time—15 s. A load-unload curve was recorded in real time, which allows the determination using the Oliver–Phare method [27], instrumental hardness H, instrumental Young E module, and allowed analysis of the work of indentation (total deformation—W_tot_, plastic deformation—W_pl_, and elastic deformation—W_sp_). Six impressions were made for all variants. A self-leveling table was used to increase the accuracy of measurements.

### 2.4. Surface Scratch Tests

The scratches were made using the Revetest device (Anton-Paar, Graz, Austria). A Berkovich indenter with a diameter of 200 µm was used in this study. The scratch test parameters were as follows: maximum load 4 N, crack length 4 mm, scratch speed 5.4 mm·min^−1^. Three scratches were made for each the tested variants. The profilographic measurements allowed to determine the surface of the groove and elevation (A_i_ and B_i_) and on their basis calculate the coefficient of wear mechanism β and the wear resistance coefficient W_β_ [28,29,30,31]:(3)$$W_{\beta }=\frac{1}{\frac{1}{n}\Sigma_{i=1}^{n}\left(\beta_{i}A_{i}\right)}\;\left[{\mathrm{mm}}^{-2}\right]$$
where:(4)$$\beta_{i}=\frac{1}{n}\overset{n}{\underset{i=1}{\sum }}\frac{A_{i}-B_{i}}{A_{i}}$$

### 2.5. Polytetrafluoroethylene-Graphite Composite Wear Tests

Tribological studies of polymers were carried out using a T-01 tribometer (manufactured by ITeE Radom, Poland). Polytetrafluoroethylene composites with 15% and 20% graphite content were tested both at the initial state and after electron beam irradiation. Three pins were prepared. The tests were performed in a pin-on-disk configuration. AISI 321 acid resistant steel (1H18N9T) and titanium grade 2 were used as discs. The surfaces of the discs were ground on 360–1200 grades to obtain a uniform surface roughness of Ra = 0.2 μm. This treatment was performed deliberately to allow faster formation of a thin film from the tested composites during the friction process, limiting the coefficient of friction. Tribological test conditions were: load 20 N (1 MPa), slip speed 10 cm/s, friction path 1 km, friction path diameter 2.4 cm. Ambient conditions in accordance with VAMAS (Versailles Project on Advanced Materials and Standards) guidelines and ASTM G-99 standard (Standard Test Method for Wear Testing with a Pin-on-Disk Apparatus) [32]. Stereometric analysis of wear traces on the pins was made using a profilograph, and then the surfaces were visualized in 3D using Talymap and Matlab software.

## 3. Results and Discussion

### 3.1. Thermal Studies

It has been determined based on DSC runs that electron-beam irradiation of PTFE with a 15% and 20% graphite additive induced a change in thermal properties of composites (Figure 1, please find the original data in Appendix A). The calculations were made on the basis of peaks shown on DSC scans illustrating the melting enanthropicity and crystallization of polyterafluoroethylene and graphite composites. On DSC scans recorded for polytetrafluoroethylene, two main peaks can be seen: the first at a temperature of about 10–40 °C and the second at a temperature of 320–340 °C. The first is related to crystal form transitions (they are most probably attributable to triclinic/hexagonal and hexagonal/pseudo-hexagonal transitions of the crystalline part of the polymer). The other is related to polymer melting. Irradiation with electron flux affected both changes in molecular weight and degree of crystallinity. The results are shown in Figure 1, Figure 2, Figure 3, Figure 4 and Figure 5.

First of all, it is clear that for a polymer with higher graphite content the heat of crystallization ΔH_c_ was approximately 20% lower in the initial state than for a composite with 15% graphite content (Figure 2). Electron beam irradiation gradually increases ΔH_c_ for both of the examined composites. The heat of crystallization did not change significantly with increase of the dose absorbed by the composites above 52 kGy.

As it results from Formula (2), the polymer composites’ heat of crystallization is closely related to their molecular weight M_n_ (Figure 3). The reduction in molecular weight observed for both PTFE composites with 15% and 20% graphite content along with the increase in the absorbed dose of electron beam is influenced by the PTFE chain disruption reactions. A polymer with higher graphite content is characterized by significantly higher average molecular weight in its initial state and, consequently, a more intensive decrease in M_n_ after electron beam irradiation.

A significant increase in the crystallinity degree of PTFE with 15% and 20% graphite as a function of electron beam irradiation is shown in Figure 4. As mentioned above, when PTFE-graphite composites absorb high doses of radiation, the molecular weight decreases due to the process of chain splitting [33]. It induces due to greater mobility and lower entanglement between polymer particles, which are much more favorable conditions for the formation of new crystallites. The authors of the papers reached similar conclusions [19,34].

With increasing absorbed radiation dose, a linear increase in melting point T_m_ of PTFE was observed for both graphite content variants (Figure 5). Furthermore, a polymer with 20% graphite content was characterized by lower temperature T_m_ in its initial condition and higher intensity of growth after electron beam irradiation. Authors of paper [34], who examined PTFE with no additives, obtained similar results after irradiating the polymer with an electron beam of energy of 1.5 MeV and 10 kGy doses.

On this basis, it can be assumed that the effect of electron beam irradiation on thermal properties will have a direct influence on PTFE’s mechanical properties and wear resistance of both polymers’ (PTFE with 15% C and PTFE with 20% C).

### 3.2. Mechanical Properties of PTFE

In spite of a lack of large differences in thermal properties for both of the examined composites subjected to irradiation, significant differences are noticeable in mechanical properties of PTFE with 20% carbon content compared to PTFE with 15% C. Electron beam irradiation of both examined composites causes changes in hardness *H* and Young’s modulus *E*, which increase proportionately to the absorbed radiation dose (Figure 6a,b). Changes in the hardness and Young’s modulus of PTFE are the consequence of the increase in the crystallinity. It was also found that a polymer with higher graphite content was characterized by 15–25% higher hardness compared to PTFE with a 15% graphite additive. As far as Young’s modulus is considered, differences between the composites are minor. However, greater increase in the elasticity modules can be observed due to the effect of the electron beam on a polymer with 20% graphite content. Both parameters (*H, E*) were subject to a considerable increase, especially in the range of 52–156 kGy, as irradiation i-multiplicity increased.

The course of load–unload curves, especially the analysis of the area under the curve allows to determine the parameters of the indentation work. The material’s resistance to deformation affects the value of the indentation work determined by the depth, volume and surface of the impressions. The total indentation work W_tot_ is the sum of the plastic deformation works W_pl_ and the elastic deformation. The area calculations were made using Matlab software.

The performed tests have shown (Figure 7a–c) that introducing additional graphite content reduces the value of total work of indentation. Further decrease in W_tot_ is observed as the electron beam irradiation dose absorbed by both of the examined polymers increases. Similar dependences occur in the case of work of plastic (W_pl_) and elastic deformation (W_el_), which reflects hardness increasing along with the growing radiation dose. Furthermore, lower susceptibility to plastic deformation (Figure 7b) and superior elastic properties (Figure 7c) can be observed in the case of a composite with higher graphite content than in PTFE with 15% graphite content.

Differences in graphite content also directly influence PTFE’s compression strength R_c_ (Figure 8). Polytetrafluoroethylene with 20% graphite content was characterized 25%–35% higher strength than PTFE with 15% C. Electron beam irradiation causes intensive growth of R_c_ of both of the examined composites up to the size of the absorbed dose (104 kGy); absorption of a dose higher than 104 kGy caused deterioration of mechanical properties of the composites. At a high radiation dose (>104 kGy), strength of a polymer decreases due to chain scission in the presence of air, consequently, R_c_ decreases as the radiation dose increases. This may also be caused by the occurrence of radiation-induced oxidation on the surface of the polymers in the presence of air [35]. During irradiation at high doses the oxide layer diffuses into the bulk of the polymer and reduces mechanical properties.

### 3.3. The Influence the Addition of Graphite and Irradiation on the Scratch Test Parameters of PTFE

Stereometric analysis of scratch traces provide information on wear mechanism β and resistance to wear W_β_ of both of the examined polymers. According to the obtained data (Figure 9a), polytetrafluoroethylene with 20% carbon content is subject to the machining mechanism to a lesser degree than PTFE with 15% C after electron beam irradiation. This means that a larger part of the furrow material undergoes plastic deformation during the scratch test and is elevated on the edge of the scratch formed. The most advantageous results for both examined composites were obtained for the absorbed dose of 104 kGy. This was corroborated by an increase in wear resistance coefficient W_β_ (Figure 9b), which causes a significant reduction in tribological wear compared to the initial material. Coefficient W_β_ increases intensively, especially in the 26–104 kGy range. However, a dose of 156 kGy causes an intensive reduction in this parameter, confirming the fact that high radiation doses lead to degradation of the plastic.

### 3.4. Wear Properties of PTFE

The reduction in tribological wear is the most important effect of radiation modification from the point of view of commercial applications of the examined composites. Figure 10 shows linear wear *W_L_* PTFE with a 15% and 20% graphite additive as a function of the absorbed irradiation dose. It is clear that already in the initial state an increase in graphite content causes a reduction in linear wear both in the case of interaction with 1H18N9T steel (by 25%) and grade 2 titanium (by 15%). Electron beam irradiation further reduces wear of both of the examined composites. The most advantageous results were obtained for polytetrafluoroethylene with 20% graphite content after absorbing a dose of 104 kGy. When steel was used as a counterpartner, the reduction in linear wear compared to the initial state was almost four-fold and more than five-fold when compared to a polymer with 15% graphite content, also in its initial state. A 2.5 fold wear reduction was observed, respectively, in the case of tribological interaction with titanium. Irradiation above a dose of 104 kGy led to polymer degradation. This effect was also visible during tribological tests. *W_L_*, increased again for a dose of 156 kGy, especially for PTFE with 20% graphite content.

Stereometric tests of the friction surface performed with a profilographometer after tribological tests confirmed the reduction in wear. Convexities and concavities, arranged as bands oriented along the motion direction, were found on the friction surface of both examined composites. During the interaction the surface becomes smooth, which smoothness increased with a higher irradiation dose, up to 104 kGy (Figure 11). Such surface morphology indicates the occurrence of lower plastic deformation as well as reduced transport of the surface material during the tribological process. Polymer degradation which occurs when applying higher radiation doses causes friction properties of PTFE to deteriorate.

## 4. Conclusions

Electron beam irradiation gradually increased crystallization heat ΔH_c_ for both examined composites. The PTFE with higher graphite content was characterized by 20% lower value of ΔH_c_ compared to PTFE+15%C in the initial state. These studies also showed that along with the pattern of absorbed radiation dose and progressive chain cleavage reactions of both PTFE-graphite composites, there is a decrease in the molecular weight of polymers.PTFE with 15 and 20% graphite content is characterized by a higher degree of crystallinity after irradiation. This is the result of changes in molecular weight and changes in the length of polymer chains after irradiation, which facilitates the crystallization process. The decreasing tendency of molecular weight of PTFE with the increasing dose of absorbed electron beam irradiation has also been shown, which can be attributed to the chain scission reaction of PTFE under increased doses.An increase in graphite content to 20% and an increase in crystallinity contributes to an increase in the compressive strength, modulus of elasticity and microhardness determined by means of microindentation. Introducing an additional quantity of graphite reduced the value of the total work of indentation *Wtot* and improved elastic properties of the composite.A decrease in susceptibility to deformation, a change in the micromechanism of wear β towards the furrow and an increase in wear resistance represented by the W_β_ coefficient was demonstrated by surface scratch tests, and their increase was affected by a change in the graphite content to 20% in the tested composites and electron beam irradiation.An increase in graphite content to 20% in the initial state caused a reduction in linear wear during interaction with both steel and titanium. Electron beam irradiation further reduced wear of both examined composites. The most advantageous results were obtained for polytetrafluoroethylene with 20% graphite content after absorbing a dose of 104 kGy. With steel as a counterpartner, the reduction in linear wear compared to the initial state was almost four-fold and more than five-fold when compared to a polymer with 15% graphite content. Profilographometric tests have shown that during the interaction the surface of polymers becomes smooth as the irradiation dose increases. This reduces transport of the material from the surface, which is a desirable phenomenon in tribological applications, especially in systems which do not require lubrication.Irradiation of PTFE with a dose of *156* kGy causes its degradation which manifests itself through degraded mechanical properties and an increase in the linear wear of both of the examined composites.Modification of PTFE through its irradiation with an electron beam may contribute to extending the life cycle of this material, e.g., in sliding components which work under heavy load conditions.

## Figures and Tables

**Figure 1 polymers-12-01676-f001:**
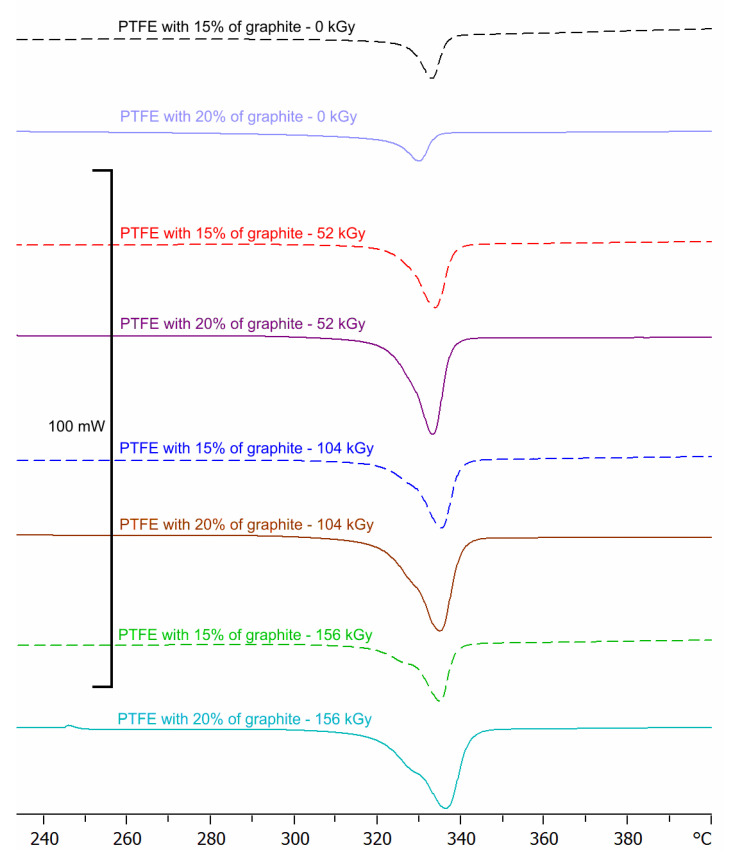
Differential scanning calorimetry (DSC) thermograms of polytetrafluoroethylene (PTFE) with 15% and 20% graphite addition.

**Figure 2 polymers-12-01676-f002:**
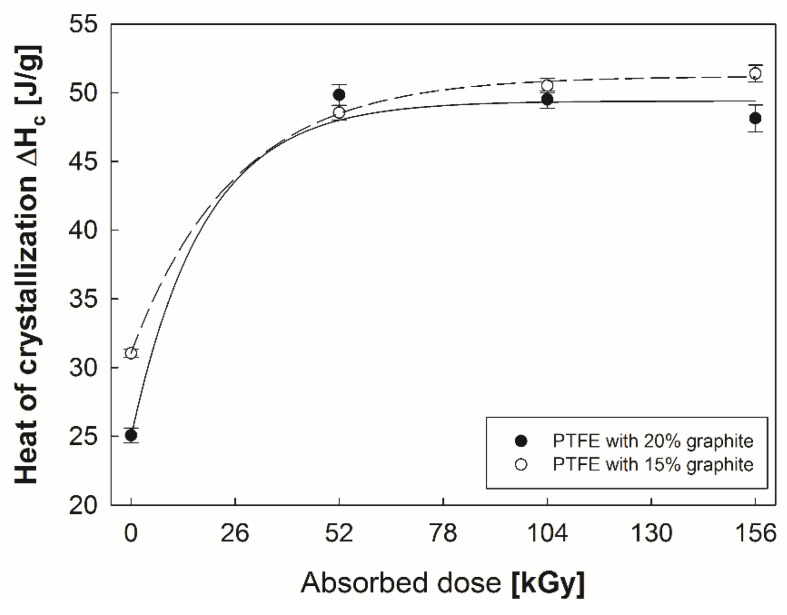
Changes in the crystallization enthalpy of PTFE with 15% and 20% graphite addition.

**Figure 3 polymers-12-01676-f003:**
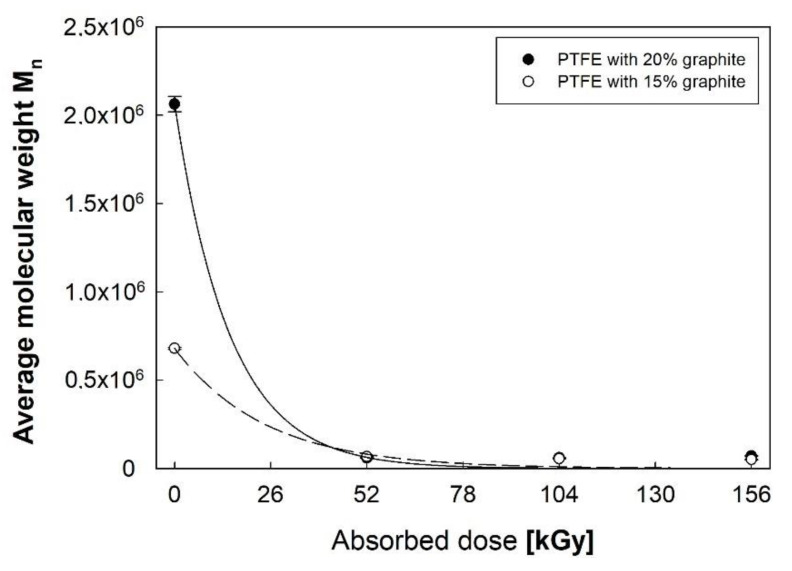
Changes in the average molecular weight of PTFE with different graphite additions.

**Figure 4 polymers-12-01676-f004:**
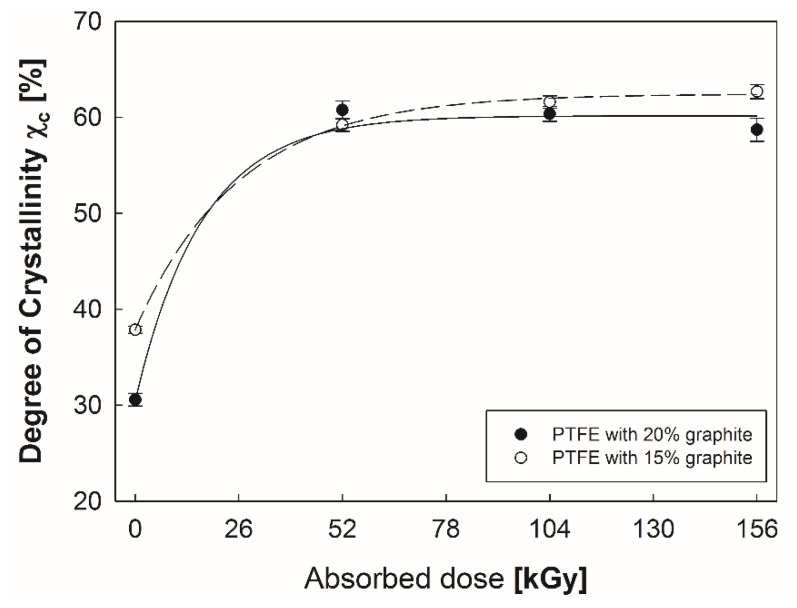
Changes in the crystallinity of PTFE with 15% and 20% graphite addition.

**Figure 5 polymers-12-01676-f005:**
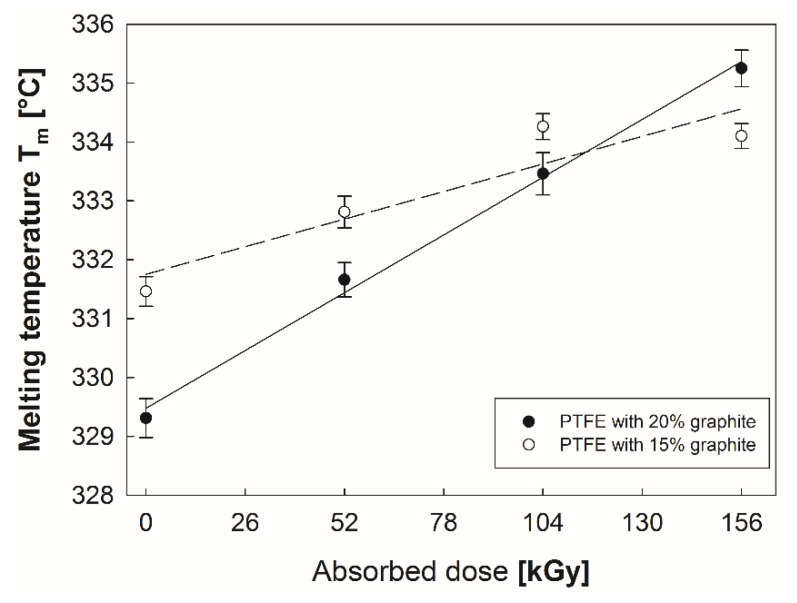
Changes in the melting point temperature of PTFE with 15% and 20% graphite addition.

**Figure 6 polymers-12-01676-f006:**
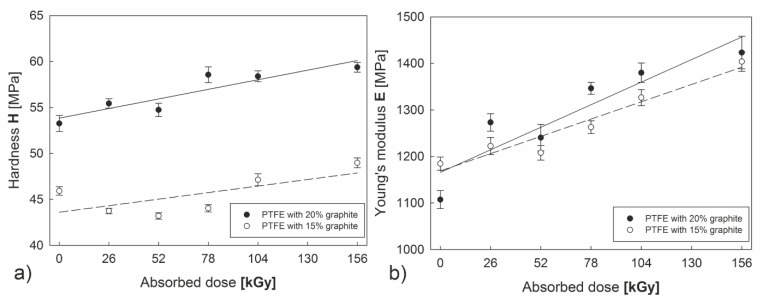
Impact of electron beam irradiation on changes in microhardness H (**a**) and modulus of elasticity E (**b**) of PTFE-graphite composites.

**Figure 7 polymers-12-01676-f007:**
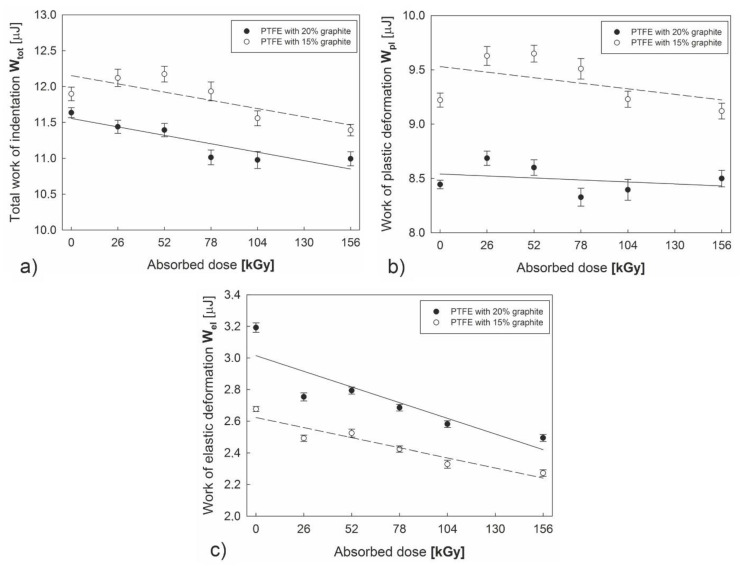
Changes in the work of indentation of PTFE with different graphite additions subjected to an absorbed dose of electron beam irradiation: the total work of indentation—W_tot_ (**a**), the work of plastic deformation—W_pl_ (**b**), and the work of elastic recovery W_el_ (**c**).

**Figure 8 polymers-12-01676-f008:**
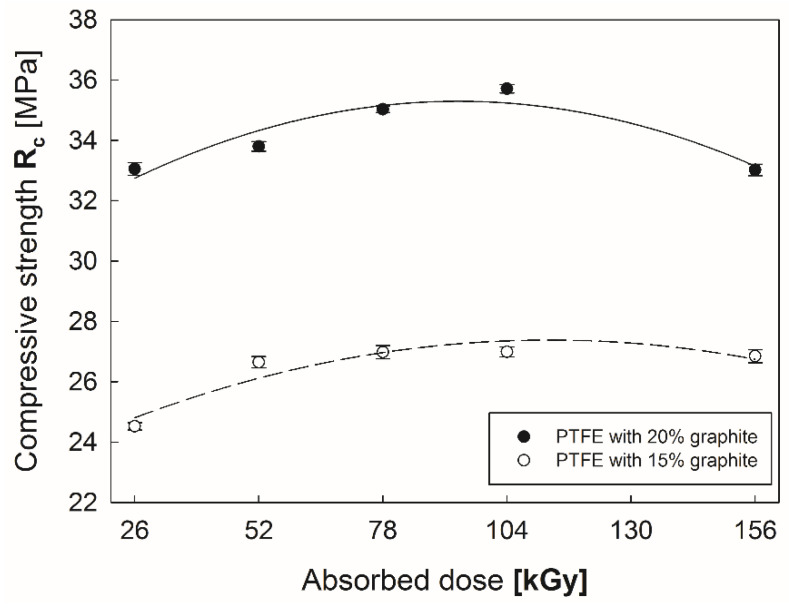
Compressive strength of PTFE with 15% and 20% graphite addition. subjected to an absorbed dose of electron beam irradiation.

**Figure 9 polymers-12-01676-f009:**
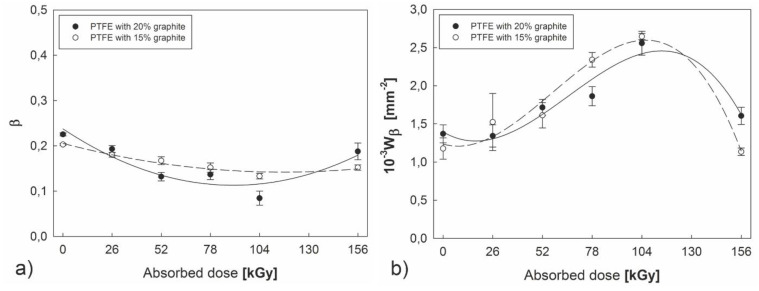
The effect of an absorbed dose of electron beam irradiation dose on the coefficient of the micromechanism of PTFE abrasive wear—β (**a**) and the abrasive wear coefficient—W_β_ (**b**).

**Figure 10 polymers-12-01676-f010:**
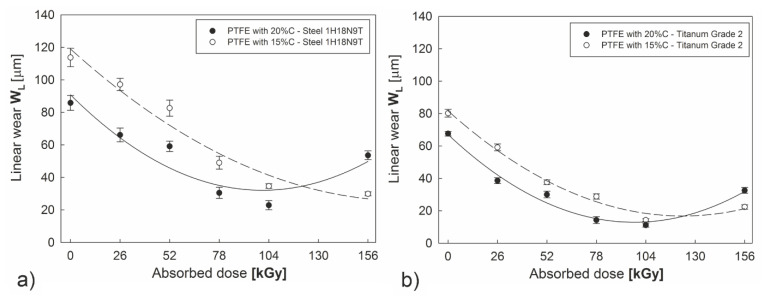
Linear wear of PTFE-graphite composites in combination with discs made of steel (**a**) and titanium grade 2 (**b**) as a function of electron beam irradiation.

**Figure 11 polymers-12-01676-f011:**
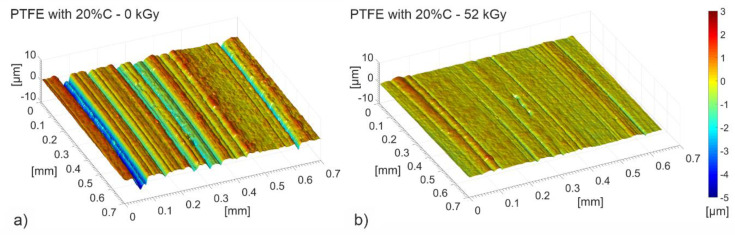
Stereometric structure of the friction surface of PTFE with a 20% graphite addition in its initial state 0 kGy (**a**) and after absorbing a dose of 52 kGy (**b**), 104 kGy (**c**), 156 kGy (**d**).

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
