# Peer review of "Tribological and Mechanical Behavior of Graphite Composites of Polytetrafluoroethylene (PTFE) Irradiated by the Electron Beam"

_polymers, 2020, doi:10.3390/polym12081676_

Round 1

Reviewer 1 Report

Line 57: Why "polytetrafluoroethylene"? the abbravation PTFE is already introduced

Line 109: Any in-homugenoeus effects having 50% compression? how to deal with inealstic deformation during unloading?

Line 115: 1N better with blank 1 N

Line 118: ist the deformation relay plastic (perfect memory) or more viscoplastic (with parts of fading mamory)?

Line 129: mm^-2 why italic?

Line 143: 20NN, better 20 N, as in Line 115

Line 211: just for (my) interest. youngs modoulus is not a eigenvalue of the stiffness tetrade, in contrast to Lamee constants.
however, ist anly knowlegde generated from the tests according to over material constants, like poissons ratio? Any information about
the development of elastic constants in view of pre-deformation? or non-symmatric behaviour in tension/compression?

Author Response

Comments and Suggestions for Authors

First of all, we would like to thank for your positive opinion regarding our article. All comments have been entered in the text.

Line 57: Why "polytetrafluoroethylene"? the abbravation PTFE is already introduced

Linia 57: As suggested by the reviewer, the abbreviation PTFE was used.

Line 109: Any in-homugenoeus effects having 50% compression? how to deal with inealstic deformation during unloading?

Linia 109: During compression tests, we did not notice such effects.

Line 115: 1N better with blank 1 N

Linia 115: Revised following the reviewer's comment. (throughout the article spacing between values and units was used)

Line 118: ist the deformation relay plastic (perfect memory) or more viscoplastic (with parts of fading mamory)?

Linia 118: The work of indentation is not perfectly plastic, its value consists of both plastic and elastic deformation work.

Line 129: mm^-2 why italic?

Linia 129: Revised following the reviewer's comment.

Line 143: 20NN, better 20 N, as in Line 115

Linia 143: Revised following the reviewer's comment.

Line 211: just for (my) interest. youngs modoulus is not a eigenvalue of the stiffness tetrade, in contrast to Lamee constants. However, ist anly knowlegde generated from the tests according to over material constants, like poissons ratio? Any information about

the development of elastic constants in view of pre-deformation? or non-symmatric behaviour in tension/compression?

 Line 211: The value of instrumental hardness and Young's modulus was calculated based on relationships formulated by Oliver-Phare, no material constants such as Poisson's ratio were calculated. The Poisson's ratio for PTFE = 0.46 was used for H and E calculations.

Reviewer 2 Report

The work is well written, the introduction focuses well on the subject.
From the point of view of the experiments, the manuscript is quite well thought out, but there are some aspects to investigate before consideration for publication. Details are provided in the following.
Some sections, in particular, should be better completed and the authors are asked to provide also original data and not only their interpreted results. For example, in Figure 1 the DSC curves show only the peak of crystallization (no endo/weight indication is present), perhaps a little less common than the usual convention (Exo: up, Endo: down). It might be a good idea to provide complete melting/crystallization cycles in the additional material for the sake of completeness.
From the standpoint of the basic principles of polymer science, it should be explained what the shoulder that appears in the lowest temperature crystallization peak is ...
How do the authors explain the shift at lower temperature observed for PTFE with 20% GR, 0 kGy?
It would be valuable to include the stress/strain curves of the indentation tests at least in the additional material because the information that can be obtained is much more than the exposed ones.

Page 9, from line 234: "At a high radiation dose (>104 kGy), the strength of a polymer decreases due to chain scission in the presence of air; consequently, Rc decreases as the radiation dose increases. This may also be caused by the occurrence of radiation-induced oxidation on the surface of the polymers in the presence of air". However, radiation exposure was carried out "The process was carried out at room temperature in a vacuum" (line 84). It should be clarified whether the vacuum conditions were effective or not before drawing conclusions (mechanical: a bulk property and scratching tests: surface properties, which are much more affected by oxidation). I recommend performing FTIR measurements to verify the presence of oxidized forms on all samples.
In Figure 11 3D scratch test images are provided only for PTFE with 20wt% graphite. Can the series also be supplied for PTFE with 15wt% graphite?

Author Response

Comments and Suggestions for Authors

The work is well written, the introduction focuses well on the subject.

From the point of view of the experiments, the manuscript is quite well thought out, but there are some aspects to investigate before consideration for publication. Details are provided in the following. Some sections, in particular, should be better completed and the authors are asked to provide also original data and not only their interpreted results.

Thank you for your positive opinion on our article and constructive comments. In the next part, we try to answer them carefully.

For example, in Figure 1 the DSC curves show only the peak of crystallization (no endo/weight indication is present), perhaps a little less common than the usual convention (Exo: up, Endo: down). It might be a good idea to provide complete melting/crystallization cycles in the additional material for the sake of completeness.

In the article, we decided to show only a part of the thermograms obtained for each studied sample for both composites, because of the readability of the drawing and because we did not want to repeat the methodology used in our previous article, regarding the PTFE-bronze composite. Of course, thermograms were recorded in a wider temperature range during heating from -40 to 400 ℃ and cooling from 400 to -40 ℃. Below is an example of PTFE with 20% carbon at baseline and after irradiation with a dose of 104 kGy:

From the standpoint of the basic principles of polymer science, it should be explained what the shoulder that appears in the lowest temperature crystallization peak is ...

On DSC recorded for polytetrafluoroethylene, two main peaks can be seen: the first at a temperature of about 10-40 ℃ and the second at a temperature of 320-340 ℃. The first is related to crystal form transitions (they are most probably attributable to triclinic / hexagonal and hexagonal / pseudo-hexagonal transitions of the crystalline part of the polymer). The other is related to polymer melting.

How do the authors explain the shift at lower temperature observed for PTFE with 20% GR, 0 kGy?

This is probably related to an increase in the graphite content (the phenomenon has not yet been fully explained). On the other hand, DSC shows that for absorbed doses of 104-156 kGy for both tested composites, the formation of an additional melting peak around 325 ° C is also visible, which is related to the process of breaking the polytetrafluoroethylene chain during the irradiation at room temperature and fragmentation of the structure crystalline polymer. Which in turn promotes the formation of new crystallites and an increase in the degree of crystallinity.

It would be valuable to include the stress/strain curves of the indentation tests at least in the additional material because the information that can be obtained is much more than the exposed ones.

Micromechanical tests were carried out at 1N load, 30s load rise time, 10s full load time, and 30s unload time fallowing ISO 14577. Load-unload curves were recorded in real time for each measurement. Examples of PTFE curves with 15 and 20% graphite content are shown below:

Page 9, from line 234: "At a high radiation dose (>104 kGy), the strength of a polymer decreases due to chain scission in the presence of air; consequently, Rc decreases as the radiation dose increases. This may also be caused by the occurrence of radiation-induced oxidation on the surface of the polymers in the presence of air". However, radiation exposure was carried out "The process was carried out at room temperature in a vacuum" (line 84). It should be clarified whether the vacuum conditions were effective or not before drawing conclusions (mechanical: a bulk property and scratching tests: surface properties, which are much more affected by oxidation). I recommend performing FTIR measurements to verify the presence of oxidized forms on all samples.

All samples subjected to irradiation, courtesy of the vacuum packaging company, were professionally protected against air. The mechanism described in these studies should therefore not be related to the oxidation of the material. However, in our previous works, we have indeed proved that, for example, in the case of polyethylene with ultra-high molecular weight UHMWPE, absorption of the electron beam dose caused the formation of free radicals and polymer cross-linking, then oxidation also occurred, which, however, managed to prevent oxidative stabilization by heating. In the case of PTFE, on the other hand, the main mechanism occurring during irradiation at room temperature is chain breakage, crosslinking occurs if the irradiation is carried out close to the melting point of the polymer, as evidenced by numerous scientific reports. We presented the FTIR study of pure polytetrafluoroethylene and bronze composite in our previous work: "Novel Organic Material Induced by Electron Beam Irradiation for Medical Application", Polymers 2020, 12, 306; doi: 10.3390 / polym12020306. They show that electron beam irradiation with polytetrafluorotethylene causes changes in the intensity of the FTIR spectrum, especially in the range of 525-800 cm-1. This phenomenon is caused by the different number of the mer units in the main chain of the polymer. The number of meric unit could be different due to the degradation process of the PTFE under irradiation of electron beam. All of the samples were exposed to dose between the range 26 - 156 kGy.

In Figure 11 3D scratch test images are provided only for PTFE with 20wt% graphite. Can the series also be supplied for PTFE with 15wt% graphite?

Placing in figure 11 the surface of mandrels only with 20% graphite content was intentional, because those with 15% content we showed in our previous article: "Radiation – chemical modification of PTFE in the presence of graphite" we wanted to avoid repetition. Below is a drawing from a published article, it shows a similar effect of reducing consumption as in the case of PTFE with 20% graphite content. However, it can be seen that PTFE consumption with a 15% C content was higher.

Round 2

Reviewer 2 Report

I have peer-reviewed the manuscript for a Journal, I think with a good reputation in the polymer field. In my opinion, the authors leak to provide original data. I think not enough to show interpreted data plots. I do not see the original data in the supplementary materials.

for example: complete melting/crystallization cycles in DSC curves, stress-strain curves, 3D scratched surfaces for PTFE with 15wt% graphite. The authors are urged to implement their revised manuscript including the data as required in the previous report or reference of previously published papers.

also the replies provided to me should be inserted in the revised manuscript. For example: Authors'reply: "On DSC recorded for polytetrafluoroethylene, two main peaks can be seen: the first at a temperature of about 10-40 ℃ and the second at a temperature of 320-340 ℃. The first is related to crystal form transitions (they are most probably attributable to triclinic / hexagonal and hexagonal / pseudo-hexagonal transitions of the crystalline part of the polymer). The other is related to polymer melting." Please, include your reply in the manuscript.

Author Response

Reviewer 2 (second round)

I have peer-reviewed the manuscript for a Journal, I think with a good reputation in the polymer field. In my opinion, the authors leak to provide original data. I think not enough to show interpreted data plots. I do not see the original data in the supplementary materials. for example: complete melting/crystallization cycles in DSC curves, stress-strain curves, 3D scratched surfaces for PTFE with 15wt% graphite. The authors are urged to implement their revised manuscript including the data as required in the previous report or reference of previously published papers.

Dear reviewer. First of all, we would like to thank you for taking the time to review our article. We're not evading from showing the original measurement. In our opinion the answers, sample drawings and data would be enough to confirm the interpretation of the results obtained. We add sample data according to the reviewer's request (DSC, micromechanical measurements, 3d surface of the wear trace, stress-strain curves). The data was exported to easy and available formats that do not require the use of specialized software only in the case of surface of the pins, these are files in the .sur format directly from the profilograph.

Also the replies provided to me should be inserted in the revised manuscript. For example: Authors'reply: "On DSC recorded for polytetrafluoroethylene, two main peaks can be seen: the first at a temperature of about 10-40 ℃ and the second at a temperature of 320-340 ℃. The first is related to crystal form transitions (they are most probably attributable to triclinic / hexagonal and hexagonal / pseudo-hexagonal transitions of the crystalline part of the polymer). The other is related to polymer melting." Please, include your reply in the manuscript.

According to the reviewer's comment, the answer was included in to the manuscript.

We hope that the attached files and previous answers will be helpful for the reviewer in assessing our work. Thank you in advance for reliable assessment of the article.

Round 3

Reviewer 2 Report

I read the replies, comments and revised manuscript provided by the authors, who unambiguously implemented their manuscript, but they fail to provide original data relevant from the point of view of the basic polymer chemistry, to be added in the supplementary material. Personally, I think the the Journal has a high reputation in the field of polymer and polymer composites, a respectable impact factor and Q1 quartile in polymer chemistry category. For these reasons I cannot accept proposals or compromises lowered compared to my requests (see my previous report). However, this is my personal opinion and you can refer to other expert reviewers in the field.

Author Response

We have implemented all suggestions and corrections required by the reviewer in our manuscript. The only remark of the reviewer at the moment is access to the original measurement files, which we sent as a supplementary file along with the previous revisions to the manuscript, and please see it in the supplementary file. In our opinion, we provided all the data collected during the research work and we do not know how we could complete the data.  

yours sincerely

Beat Knechtle